# Atlantic meridional overturning circulation increases flood risk along the United States southeast coast

Denis L. Volkov [1,2] ✉, Kate Zhang [3], William E. Johns[4], Joshua K. Willis [5], Will Hobbs [6,7], Marlos Goes [1,2], Hong Zhang[5] & Dimitris Menemenlis[5]

The system of oceanic flows constituting the Atlantic Meridional Overturning Circulation (AMOC) moves heat and other properties to the subpolar North Atlantic, controlling regional climate, weather, sea levels, and ecosystems. Climate models suggest a potential AMOC slowdown towards the end of this century due to anthropogenic forcing, accelerating coastal sea level rise along the western boundary and dramatically increasing flood risk. While direct observations of the AMOC are still too short to infer long-term trends, we show here that the AMOC-induced changes in gyre-scale heat content, super-imposed on the global mean sea level rise, are already influencing the frequency of floods along the United States southeastern seaboard. We find that ocean heat convergence, being the primary driver for interannual sea level changes in the subtropical North Atlantic, has led to an exceptional gyre-scale warming and associated dynamic sea level rise since 2010, accounting for 30-50% of flood days in 2015-2020.

Sea level rise is among the most challenging consequences of global warming. Since 1900, the Global Mean Sea Level (GMSL) has swelled by about 20 cm[1], with the average rate over the past 30 years of 3.5 mm yr$^{-1}$ (ref. 2). As the melting of the terrestrial ice and ocean warming intensify, sea level rise is accelerating and projected to rise by over half a meter by the end of this century[3–5]. Low-lying coastal regions, including areas along the United States (U.S.) eastern seaboard and Gulf of Mexico, are the most vulnerable to sea level rise. Coastal flooding mainly occurs due to large synoptic sea level fluctuations, driven by atmospheric pressure and wind forcing, such as tropical storms and hurricanes, often superimposed on spring tides, lasting from a few hours to several days. The rising background coastal water levels allow seawater to reach further inland during storms and surges. This causes wetland flooding, land loss and erosion, salinization of the aquifer and agricultural soil, and devastation of coastal habitats and ecosystems. It also endangers near-shore infrastructure and forces populations to migrate to higher ground[6–9].

While the GMSL is rising, ocean and atmosphere dynamics make sea level changes spatially variable, with some regions where the sea level rising much faster than the global mean[10–12]. The U.S. east coast has been identified as a hotspot for accelerated sea level rise in the North Atlantic[13,14]. For example, the rates of coastal sea level rise south of Cape Hatteras in 2010–2015 were up to 5 times greater than the global average, while negative trends were observed north of Cape Hatteras[15,16]. In prior decades, however, accelerated sea level rise was observed north of Cape Hatteras[17–19]. A recent shift in the hotspot of the accelerated sea level rise that occurred in 2010 has been attributed to changes in the Gulf Stream strength and position[20]. On a multi-decadal timescale, it has been shown that the large-scale sea level changes to the north and to the south of Cape Hatteras were coherent before 1990 and incoherent afterwards[21]. This regional acceleration and spatial variation

[1]Cooperative Institute for Marine and Atmospheric Studies, University of Miami, Miami, FL, USA. [2]NOAA Atlantic Oceanographic and Meteorological Laboratory, Miami, FL, USA. [3]Joint Institute for Regional Earth System Science and Engineering, University of California Los Angeles, Los Angeles, CA, USA. [4]Rosenstiel School of Marine, Atmospheric, and Earth Science, University of Miami, Miami, FL, USA. [5]Jet Propulsion Laboratory, California Institute of Technology, Pasadena, CA, USA. [6]Australian Research Council Centre of Excellence for Climate Extremes, Sydney, NSW, Australia. [7]Australian Antarctic Program Partnership, Institute for Marine and Antarctic Studies, University of Tasmania, Hobart, TAS, Australia. ✉e-mail: denis.volkov@noaa.gov

of sea level rise has been attributed to different processes, such as longshore wind forcing[22–24], atmospheric pressure loading[25], vertical land motions[26], weakening of the Gulf Stream[27–32], warming of the Gulf Stream and the entire subtropical gyre[15,33], the combined influence of external forcing and internal climate variability[34], and slowdown of the Atlantic Meridional Overturning Circulation (AMOC)[35–38].

The regional sea level variations driven by the ocean and atmospheric dynamics, corrected for the inverted barometer effect and not directly related to ice loss or net heat absorbed by the ocean, are termed dynamic sea level changes. State-of-the-art climate models project a decline in the AMOC towards the end of the century, which, as a consequence of geostrophic balance, would be accompanied by a dynamic sea level rise along the western boundary of the North Atlantic[35–38]. Higher sea levels are expected to dramatically increase the risk of coastal flooding[9,39]. While century-long proxy data indicates that the AMOC may already be slowing down[40], the direct observations of the AMOC are still too short to confirm this centennial trend[41], and they mainly showcase interannual-to-decadal variations[42–45].

The regional dynamic sea level changes and the GMSL rise superimpose on each other and provide background conditions for large-amplitude synoptic and tidal fluctuations. In addition, land subsidence with an average rate of about 1 mm yr$^{-1}$ is a sizable contributor to the accelerated sea level rise along the U.S. East coast[15,46–49]. In low-lying coastal regions, an increase of even a few centimeters in the background sea level can break the regional flooding thresholds and lead to coastal inundation. Furthermore, as the GMSL continues to rise, the impact of the regional dynamic sea level changes on coastal inundation is increasing. In this study, by analyzing Sea Surface Height (SSH) measurements from satellite altimetry and tide gauge records, we quantify how the gyre-scale dynamic SSH changes in the subtropical North Atlantic are already influencing the frequency of flooding events along the U.S. southeast coast, including the Gulf of Mexico. We show that these sea-level changes are mainly due to changes in Oceanic Heat Content (OHC), and we use an ocean-circulation model constrained by observations to demonstrate that heat advection is the dominant term in the subtropical North Atlantic heat budget. We thus establish a link between the AMOC-driven gyre-scale ocean heat convergence and

coastal-flood risk. The key hypothesis behind this study is that the AMOC plays an important role in the development of anomalous large-scale OHC and sea level patterns, which, in turn, affect the coastal sea level and the frequency of floods.

## Results

### The North Atlantic SSH tripole

In the North Atlantic, the interannual-to-decadal dynamic SSH variability (with the GMSL subtracted) is characterized by a large-scale tripole pattern, known as the North Atlantic SSH tripole, in which the subtropical ocean gyre varies out-of-phase with both the subpolar gyre and the tropics[33,50] (Fig. 1a). The tripole is defined as the leading Empirical Orthogonal Function ($EOF_1$) of the low-pass filtered dynamic SSH with a cutoff period of 1.5 years (Methods). The first Principal Component ($PC_1$) depicts the time evolution of the tripole, characterized by an overall SSH decrease in the subtropical gyre in 1993–2010 and a rapid SSH rise in 2010–2015 (Fig. 1c). The opposite tendencies were observed in the subpolar gyre and in the tropics.

It has been reported that the North Atlantic SSH tripole is correlated with the gyre-scale oceanic heat convergence closely linked to the AMOC and with the North Atlantic Oscillation[33], suggesting that the tripole results from the adjustment of the large-scale ocean circulation to variable surface buoyancy and wind forcing. The upper 2000-m OHC, represented by the thermosteric SSH derived from temperature measurements (Methods), also exhibits an $EOF_1$ pattern similar to the one shown in Fig. 1a, and its $PC_1$ is well correlated with the $PC_1$ of altimetric SSH ($r = 0.96$; compare red and black curves in Fig. 1c). This means that the tripole is a good proxy for the OHC interannual variability in the North Atlantic. The potential temperature averaged over the area 30°W–70°W and 25°N–40°N (black rectangle in Fig. 1a), characteristic for the subtropical gyre, illustrates the tripole-related changes, with exceptionally strong warming of nearly 1 °C at the surface in 2010–2015 and a deep extension down to about 1500-m depth (Fig. 1d). This tripole-related ocean warming extended all the way towards the Florida Straits, where depths barely exceed 700-m, which led to an accelerated sea level rise along Florida east coast[15].

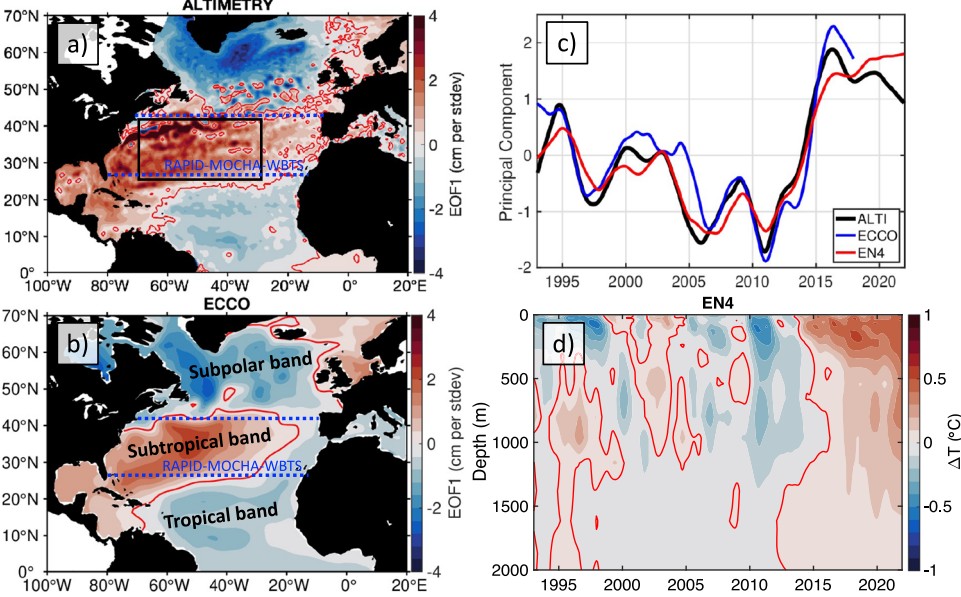

**Fig. 1 | The North Atlantic Sea Surface Height (SSH) tripole.** The spatial patterns of the leading Empirical Orthogonal Functions ($EOF_1$) of the low-pass filtered SSH anomalies in **a** satellite altimetry data and **b** in the ECCO model; **c** the temporal evolutions of $EOF_1$ patterns (Principal Components, $PC_1$) for satellite altimetry SSH anomalies (black), for SSH anomalies in the ECCO model (blue), and for thermosteric SSH anomalies in EN4 data (red). **d** The time-depth diagram of the upper 2000-m potential temperature averaged over the area 30°W–70°W and 25°N–40°N outlined by the black rectangle in panel **a**. The blue dashed lines in **a**, **b** show the 26.5°N (RAPID-MOCHA-WBTS) and 41°N transects, across which the observational estimates of the meridional heat transports are available.

## Relationship between the SSH tripole and the AMOC

In order to establish a robust dynamic relationship between the tripole and AMOC, we employed ocean state estimates from Estimating the Circulation and Climate of the Ocean Version 4 Release 4 model (ECCO-V4r4, hereinafter ECCO), a solution that is constrained by selected satellite and in situ data (Methods)[51]. An advantage of using output from a data-constrained model is that it allows exact computations of regional oceanic heat budgets while being consistent with observations. Despite a rather coarse horizontal resolution (nominal 1° horizontal grid spacing), the ECCO solution realistically simulates the large-scale interannual SSH variability in the North Atlantic. The simulated tripole (EOF$_1$) pattern is similar to the observed one (compare Fig. 1b, a) and the correlation between the PC$_1$ of the simulated SSH and the PC$_1$ of the observed SSH is 0.92 (compare blue and black curves in Fig. 1c).

As the tripole is strongly related to the OHC, its variability is partly driven by heat advection. Luckily, the northern and southern boundaries of the subtropical band of the tripole lie close to the latitudes across which the observational estimates of meridional heat transport (MHT) are available (blue dashed lines in Fig. 1a, b). These are the estimates based on measurements collected by the RAPID-Meridional Overturning Circulation and Heat-flux Array−Western Boundary Time Series (RAPID-MOCHA-WBTS) moored array at about 26°N (referred hereafter as RAPID)[52,53] and the estimates based on the combination of satellite altimetry and Argo data at 41°N from Hobbs and Willis (hereinafter HW12)[54–56]. The monthly MHT anomalies at both latitudes from ECCO and from observations reasonably agree with each other, with the correlation coefficients between ECCO and observational estimates being 0.53 at 41°N (Fig. 2a) and 0.86 at 26°N (Fig. 2b). The

significant difference between the correlation coefficients is likely due to the different methodologies used to derive the HW12 and RAPID estimates rather than the different performance of ECCO at the two latitudes. The time-mean MHT at 41°N for the concurrent ECCO and HW12 data (2002−2017) is 0.43 and 0.41 PW, respectively. The time-mean MHT estimates at 26°N for the concurrent records (2004−2017) are significantly different: 0.83 PW for ECCO and 1.19 PW for the RAPID data. This means that ECCO underestimates ocean heat convergence in the North Atlantic subtropical gyre by about 50%. Nevertheless, because ECCO realistically simulates the tripole and the time-variable part of the MHT, it is reasonable to assume that the time-variable heat budget components in ECCO are robust.

The full-depth, basin-wide oceanic heat budget is assessed between the latitudes of 26°N and 41°N, which bound the largest part of the subtropical band of the tripole (Fig. 2c). In ECCO, the time-change of the volume-integrated OHC between the two latitudes (black curve in Fig. 2c) is the sum of (i) heat advection (ocean heat convergence), which is the difference between the MHT at 26°N and 41°N (red curve in Fig. 2c), (ii) the volume-integrated forcing term that includes surface heat exchanges and geothermal forcing (blue curve in Fig. 2c), and (iii) the one due to diffusion terms (green curve in Fig. 2c) (Methods). The heat convergence computed from the model and from RAPID and HW12 data are significantly correlated ($r = 0.6$; solid and dotted red curves in Fig. 2c, respectively), providing more confidence in the time-variable parts of the simulated heat budget terms. As these terms illustrate, the subtropical band of the tripole gains heat due to the meridional oceanic heat convergence (positive advection term), and it loses heat to the atmosphere due to diffusion. What is important to note for the objectives of this study is that the MHT convergence is

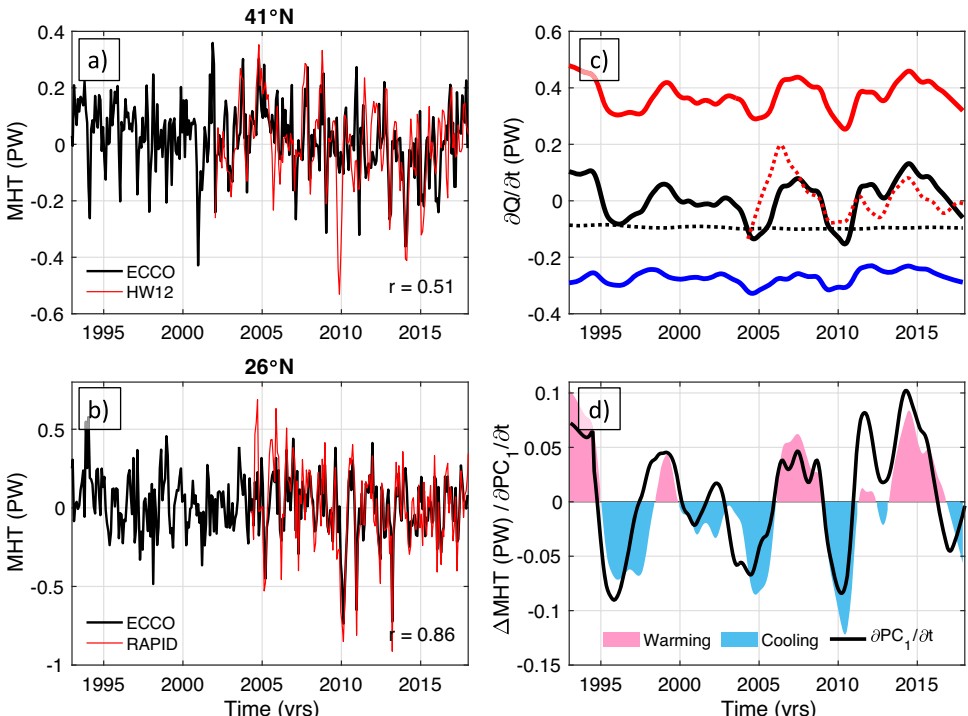

**Fig. 2 | Oceanic heat budget in the subtropical North Atlantic. a** The time series of the meridional heat transport anomalies relative to the common time intervals at 41°N (black) in the ECCO model and (red) from Hobbs and Willis, 2012 (HW12); **b** the time series of the meridional heat transports (black) in the ECCO model at 26°N and (red) from RAPID observations at about 26°N; **c** the full-depth ocean heat budget between 26°N and 41°N: (black) the time-change in Oceanic Heat Content, (red) heat advection, (blue) the volume-weighted averaged forcing term that includes surface forcing, penetrated shortwave radiation, and geothermal forcing,

(dotted black) the diffusion term; the dotted red curve shows the heat divergence anomaly from observations (difference between the Meridional Heat Transport anomalies at 26.5°N and at 41°N); **d** the oceanic heat convergence between 26°N and 41°N in the ECCO model (red and blue shading indicating warming and cooling between the two latitudes, respectively) and the time-derivative of the leading Principal Component (PC$_1$) of the low-pass filtered Sea Surface Height from satellite altimetry ($\partial$PC$_1$/$\partial$t; black curve).

the primary driver for the interannual variability of OHC in the subtropical band of the tripole. The correlation between the advection term and the time-change of OHC is 0.96, and the standard deviations of these terms are 0.06 and 0.07 PW, respectively. The correlation between the forcing term and the time-change of OHC is 0.69, but the amplitude of the former is only 0.02 PW. As expected, the contribution of the diffusion term to the interannual variability of OHC is negligible.

## The AMOC and coastal sea level

The MHT and the AMOC estimates are strongly correlated at 26°N and 41°N ($r = 0.96$ at 26°N and $r = 0.85$ at 41°N), meaning that the MHT variability is mainly due to the variability of the meridional velocity rather than temperature. Furthermore, the MHT variability at the two locations is dominated by the overturning circulation and not by the horizontal gyre circulation[53,54]. Therefore, the MHT convergence anomalies between the two latitudes mostly denote the AMOC-driven OHC tendencies (warming and cooling) in the subtropical band of the tripole (red and blue shading in Fig. 2d). These tendencies are well correlated with the time-derivative of the PC$_1$ of the altimetric SSH ($r = 0.82$; a black curve in Fig. 2d). While the time-integral of the MHT convergence determines the gyre-scale SSH and OHC, it also exerts influence on coastal sea level. For example, it has recently been shown that the tripole explains up to 60–80% of the interannual coastal sea level variance along the U.S. southeastern seaboard[33]. The amplitudes of the tripole-related coastal sea level changes, obtained by regressing tide gauge records on the PC$_1$ of the altimetric SSH (Methods), are small (0–2 cm) at the tide gauges situated to the north of Cape Hatteras, but they sometimes exceed 10 cm at the tide gauges located to the south of Cape Hatteras and in the Gulf of Mexico (Fig. 3a). The small amplitudes of the tripole-related SSH changes at the former tide gauges are probably due to the proximity to the boundary

between the subtropical and the subpolar bands of the tripole. Most importantly, the relatively large amplitudes along the South Atlantic Bight and Gulf coasts are comparable to the absolute GMSL rise since 1993 (see insert in Fig. 3a). This means that along the U.S. Southeast and Gulf coasts the impact of the tripole, i.e., the gyre-scale dynamic SSH changes, on the frequency of coastal inundations is equal or sometimes even greater than the impact of the GMSL rise.

Displayed in Fig. 3b are the probability density functions for daily highest water levels relative to the 1983–2001 mean higher high water (MHHW) tidal datum at several characteristic tide gauges to the north (Boston and Atlantic City) and to the south (Virginia Key and Galveston Pier) of Cape Hatteras, calculated over 5-year time intervals. Due to the GMSL rise from 1995–2019, the mean values of the quasi-Gaussian distributions have been shifting towards higher water levels, thus increasing the probability of water levels breaking the minor flood thresholds[9] (vertical orange lines in Fig. 3b). What is interesting to note is that south of Cape Hatteras the probability density functions for the years of 2015–2019 (red curves in Fig. 3b, lower plots) clearly stand out as exhibiting the largest shift relative to the previous (2010–2014) time interval (green curves in Fig. 3b, lower plots). We demonstrate below that this shift is partly due to the North Atlantic SSH tripole, the impact of which has become more prominent. Since 2014, the tripole has been in its positive phase, characterized by positive SSH anomalies in the subtropical gyre and negative SSH anomalies in both the subpolar gyre and in the tropics. The maximum SSH in the subtropical gyre was observed in 2015-2016. Because the amplitude of this tripole-driven change along the U.S. Southeast and Gulf coasts is similar to the absolute GMSL rise since 1993, it is reasonable to expect that both the GMSL rise and the tripole have been equally significant preconditioning factors for coastal inundation events in recent years.

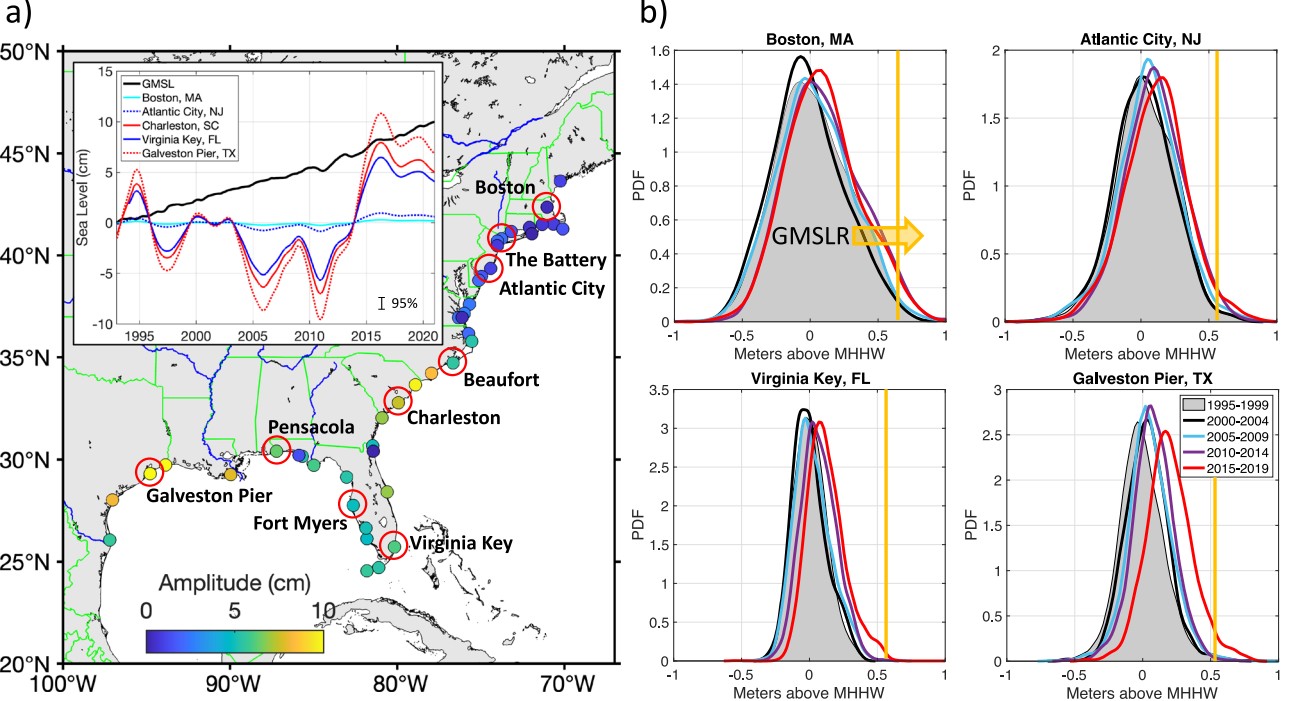

**Fig. 3 | The North Atlantic Sea Surface Height (SSH) tripole and coastal sea level. a** The locations of tide gauges and the amplitude of the tripole-related SSH changes at these tide gauges (colored circles). An insert shows (black) the Global Mean Sea Level (GMSL) change and (other color curves) the tripole-related SSH time series at several tide gauges. The vertical error bar in the insert shows the 95% confidence interval for regression. **b** Probability density functions for daily highest water levels relative to 1983–2001 mean higher high water (MHHW) tidal datum at

Boston (MA), Atlantic City (NJ), Virginia Key (FL), and Galveston Pier (TX) tide gauges over the following time intervals: (gray shading) 1995-1999, (black) 2000–2004, (blue) 2005-2009, (purple) 2010–2014, (red) 2015–2019. The vertical orange lines indicate the NOAA minor flood thresholds published in a NOAA Technical Report[9]. The orange arrow points in the direction of the shift of probability density functions due to the GMSL rise (GMSLR).

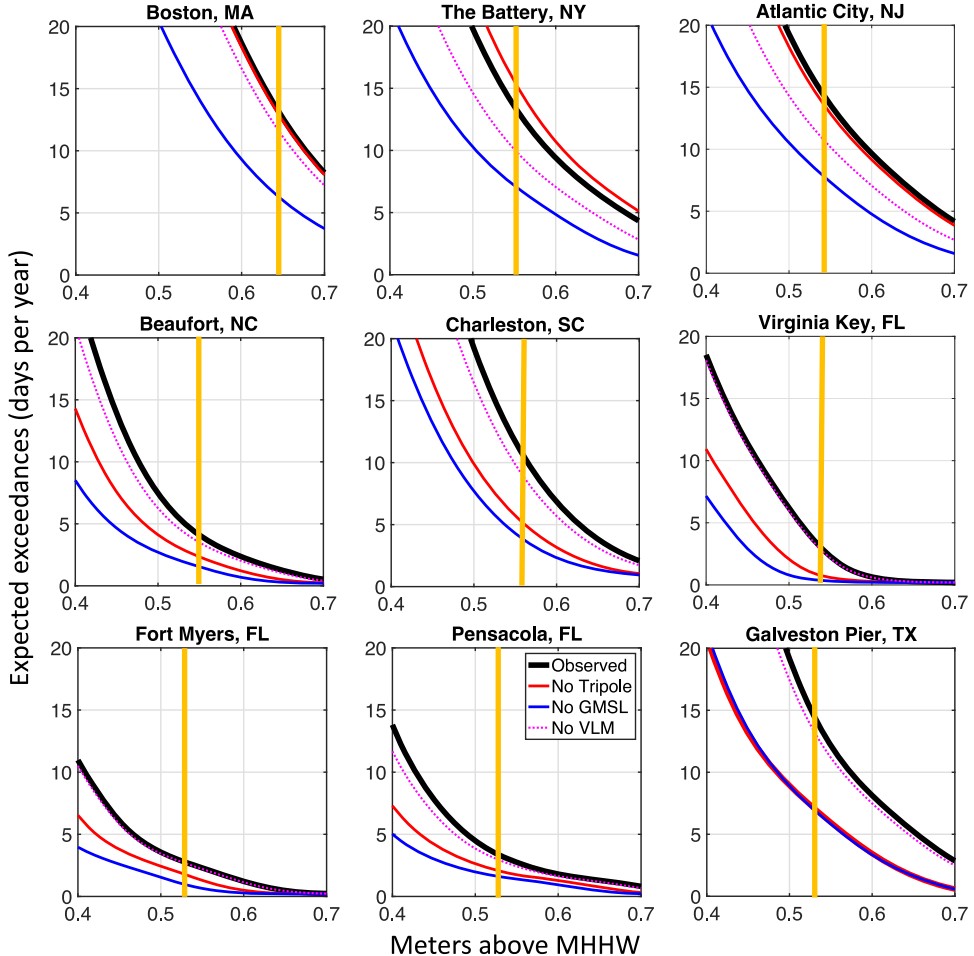

**Fig. 4 | Annual expected exceedances for daily highest water levels in 2015–2019 relative to 1983–2001 mean higher high water (MHHW) tidal datum at several tide gauges along the U.S. East and the Gulf of Mexico coasts.** (Black) Expected exceedances for the actual (Observed) sea level records, (red) for the records with the tripole-related sea level subtracted (No Tripole), (blue) for the records with the Global Mean Sea Level subtracted (No GMSL), and (dotted magenta) for the records corrected for vertical land motion (No VLM). The vertical orange lines indicate the NOAA Minor flood thresholds published in a NOAA Technical Report[9]. The annual expected exceedances are shown for tide gauges highlighted by red circles in Fig. 3a.

## Changes in the frequency of coastal inundations

The annual expected exceedances for daily highest water levels in 2015–2019 at several tide gauges along the U.S. east and Gulf coasts show the impact of vertical land motions, the GMSL rise, and the tripole on the frequency of floods (Fig. 4). A flood day occurs when the water level exceeds the minor flood threshold (vertical orange lines in Fig. 4) for at least an hour. Vertical land motions in the region are generally represented by land subsidence[46–49], which increases the frequency of floods. When the tide gauge records are corrected for land subsidence since 1993, the number of flood days in 2015–2019 reduces at nearly all analyzed locations (compare black and dotted magenta curves in Fig. 4). Because the rates of land subsidence are stronger north of Cape Hatteras (1–2 mm yr⁻¹) and weaker along the South Atlantic Bight and Gulf coasts (<1 mm yr⁻¹), this hypothetical reduction is significant (up to 25%) in the former region and relatively small in the latter region. As expected, the strongest contribution to the increasing frequency of floods comes from the GMSL rise. If the GMSL rise since 1993 were removed from the tide gauge records, there would be about a two-fold or greater decrease in the annual number of flood days in 2015–2019 at almost all locations (compare black and blue curves in Fig. 4).

But what about the impact of the gyre-scale ocean heat content variability, represented by the North Atlantic SSH tripole? If the tripole-related SSH changes were removed from tide gauge records, the annual number of flood days north of Cape Hatteras would not change significantly (compare black and red curves in the upper panels of Fig. 4), which means that the impact of the tripole on the frequency of floods here is small or negligible. At some tide gauges to the north of Cape Hatteras, like at The Battery, NY (middle plot in the upper panels of Fig. 4), the removal of the tripole-related changes may even result in a small increase in flood events. This is because this stretch of the coastline is close to the boundary between the subtropical and the subpolar bands of the tripole (Fig. 1a), and some tide gauges there may belong to the latter. On the other hand, the removal of the tripole-related changes from tide gauge records to the south of Cape Hatteras would reduce the annual number of flood days in 2015–2019 by 30–50% (compare black and red curves in the middle and lower panels of Fig. 4). The impacts of the tripole and the GMSL rise on the frequency of floods along the U.S. Southeast and Gulf coasts in recent years have been comparable. At some locations, like Galveston Pier, TX, the hypothetical reduction in flood events in the absence of the GMSL rise and in the absence of the tripole variability are the same.

Generally, the stretch of the coastline north of Cape Hatteras has been more vulnerable to flooding than the U.S. Southeast and the Gulf coasts (Fig. 5a). Due to the GMSL rise, along the U.S. Northeast coast, the number of flood days per year has been increasing, reaching a maximum in 2018. On the other hand, the frequency of floods south of

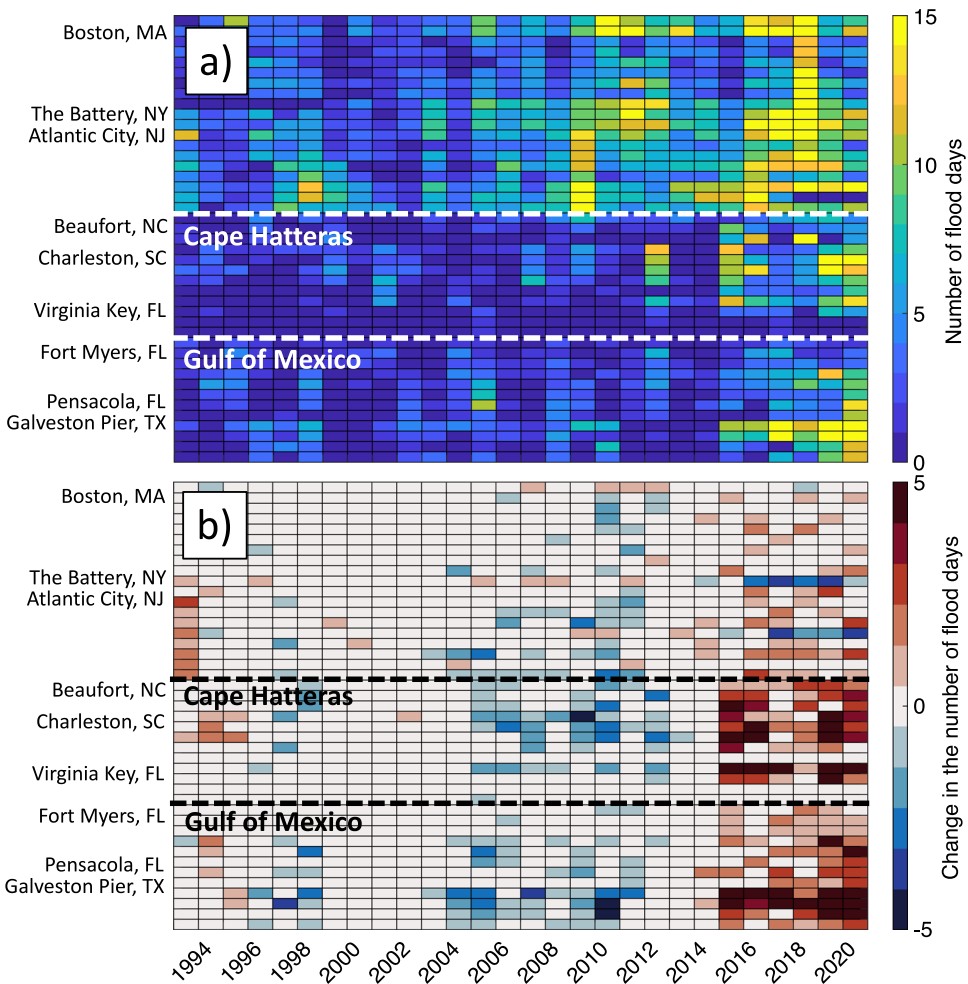

**Fig. 5 | Impact of the North Atlantic Sea Surface Height (SSH) tripole on coastal-flood risk. a** The number of flood days per year between 1993 and 2020 at each tide gauge used in this study (indicated on the vertical axis). A flood day is defined as the day when the water level exceeds the Minor flood threshold for 1 h or more. **b** The difference between the actual number of flood days and the number of flood days after subtracting the tripole-related sea level from tide gauge records. The positive/negative values indicate years and tide gauges, when and where the North Atlantic SSH tripole was increasing/decreasing the frequency of floods. The horizontal dashed lines show the approximate locations of Cape Hatteras and the boundary between the U.S. Atlantic and Gulf coasts.

Cape Hatteras, including the Gulf coast, has been rather stable until 2015. Since then, the number of floods per year increased substantially and became comparable to the number of floods north of Cape Hatteras. It appears that this change can be largely attributed to the tripole. Indeed, south of Cape Hatteras, the tripole-related changes are responsible for up to 5 flood days per year in 2015–2020 (Fig. 5b), constituting 30–50% of the total number of floods (compare Fig. 5a, b). Because the tripole was characterized by an SSH decrease in the subtropical gyre in 1995–2010, it was reducing the number of floods during this period (blue shading in Fig. 5b), which apparently compensated for the impact of the GMSL rise.

## Discussion

The idealized relationship between sea level changes along the U.S. east coast and the AMOC is determined by the zonally integrated geostrophic balance: a stronger northward transport is associated with a lower coastal sea level. Although this antiphase relationship is generally reproduced by the current generation of large-scale climate and ocean models, it is less evident in observations, probably due to the shortness of observational records and to the impact of local processes not resolved by the models[37]. In this study, we have followed an alternative concept and approach by establishing a connection between dynamic sea level changes, characterized by the North

Atlantic SSH tripole[33,50], and the AMOC-driven heat convergence. We have focused on the subtropical band of the tripole, essentially, the North Atlantic subtropical gyre, where observational estimates of the AMOC at 26°N (RAPID array)[52,53] and at 41°N (combination of satellite altimetry and Argo measurements)[54] are available. The relationship between OHC and dynamic SSH in the subtropical gyre and the AMOC has been established by calculating the regional heat budget between 26°N and 41°N in the ECCO-V4r4 state estimate. The ECCO MHT variability at these latitudes agrees well with observational estimates, which provides confidence in the robustness of the simulated heat budget components. Our results demonstrate that the interannual variability of OHC in the subtropical band of the tripole is primarily driven by MHT convergence, while the contribution of the net surface heat flux is of secondary importance.

The North Atlantic SSH tripole, being a gyre-scale variability pattern, exerts its influence on sea level along the U.S. East coast[33]. By regressing tide gauge measurements on the temporal evolution of the tripole, we have shown that the tripole is most influential south of Cape Hatteras and in the Gulf of Mexico, where the tripole-related coastal sea level changes can reach amplitudes of about 10 cm, which is close to the GMSL rise magnitude over the last 30 years. Together with the GMSL rise and the seasonal SSH variability, the tripole-related interannual-to-decadal SSH changes provide background conditions

for larger-amplitude synoptic and tidal fluctuations. This study provides observational evidence that tripole-related SSH changes impact the frequency of floods. A particularly strong positive SSH anomaly has been observed in the subtropical band of the tripole since 2015, mainly resulting from the AMOC-driven gyre-scale heat convergence. During this time, the frequency of floods along the South Atlantic Bight and the Gulf coasts increased dramatically, which is partly due to the impact of tripole-related SSH changes. If the tripole variability were absent, the frequency of floods since 2015 would be 30–50% less than present. We have shown that the impact of the tripole on coastal-flood risk in the region since 2015 is comparable to the impact of the GMSL rise since 1993. It should be noted that the influence of tripole-related changes has increased over time as the GMSL is steadily rising. During 1993–2010, the tripole was characterized by a general reduction of SSH in the subtropical gyre, which compensated for the GMSL rise and resulted in a relatively stable flood incidence. It is reasonable to expect that, with the continued GMSL rise, multi-year periods where the tripole-related variability greatly enhances the frequency of flooding should be expected.

The results of this study highlight the importance of accounting for natural, large-scale sea level variability in order to improve coastal sea level projections and to better assess coastal-flood risk. Because its mechanisms are not yet fully understood and because it is difficult to predict, this natural variability is often neglected in coastal-flood modeling and projections. For the South Atlantic Bight and Gulf coasts, we have established a strong link between coastal sea level, the associated flood frequency, and gyre-scale dynamic SSH and OHC variability, which are largely controlled by AMOC-driven ocean heat convergence. Because basin-wide SSH and OHC are proportional to the time-integral of MHT convergence, measuring MHT provides the potential for multi-year SSH predictability. This makes continued satellite and Argo observations highly valuable and illustrates the particular importance of the RAPID section, which is nearly aligned with the southern boundary of the subtropical band of the tripole. Together, these three observing systems are particularly valuable for predicting coastal sea levels and making flood-risk projections.

## Methods

Sea level has been continuously observed by altimetry satellites with near-global coverage since the end of 1992. Here, we used the monthly maps of SSH anomalies for the period from January 1993 to December 2020 processed and distributed by the Copernicus Marine and Environment Monitoring Service (CMEMS; http://marine.copernicus.eu). The SSH anomalies are computed with respect to a 20-year (1993–2012) mean sea surface. To focus on the dynamic SSH variability, we subtracted the GMSL from SSH anomalies at each grid point. We also used the Met Office Hadley Centre EN4 monthly gridded temperature profiles to compute thermosteric (due to temperature variations only) SSH fields (Fig. 1c) and to illustrate subsurface temperature changes in the subtropical North Atlantic (Fig. 1d). The seasonal cycle was computed by least squares fit of both the annual and the semi-annual harmonics and subtracted from the data.

The North Atlantic SSH tripole is defined as the leading Empirical Orthogonal Functions (EOF$_1$) mode of the low-pass filtered dynamic SSH with a cutoff period of 1.5 years. The EOF analysis is performed over the North Atlantic domain, 0°–70°N and 100°W–20°E, yielding the spatial pattern of SSH change (EOF$_1$; Fig. 1a) and its temporal evolution (PC$_1$; Fig. 1c). The EOF$_1$ explains 28% of the interannual SSH variance. It is shown as a regression map (Fig. 1a, b) obtained by projecting SSH data onto the standardized (divided by standard deviation) PC$_1$ time series so that the regression coefficients are in centimeters (local SSH change) per 1 standard deviation change of PC$_1$. The EOF$_1$ mode explains the majority of the interannual SSH variance in the northwestern North Atlantic, in the subtropical gyre, and—most

importantly for the objectives of this study—along the South Atlantic Bight, the Gulf of Mexico, and the Caribbean coasts[33]. The second EOF mode explains 14% of the interannual SSH variance, but it mostly accounts for the interannual SSH variability in the northeastern part of the subpolar North Atlantic[50]. Therefore, this mode is not considered in this study. The correlation coefficients mentioned in the paper are significant at 95% confidence.

The Estimating the Circulation and Climate of the Ocean Version 4 Release 4 (ECCO-V4r4) ocean state estimate was used to estimate the regional ocean heat budget between 26°N and 41°N. The ECCO consortium aims to produce accurate, physically consistent, time-evolving estimates of ocean circulation by constraining the Massachusetts Institute of Technology general circulation model (MITgcm) with the most available in situ and satellite observations[51]. The adjoint method is used to iteratively minimize the squared sum of weighted model-data misfits and to adjust a set of model control parameters[57]. The ECCO-V4r4 solution covers the period from January 1993 to December 2017, and it is available at https://ecco-group.org/. An EOF analysis was also applied to de-seasoned, low-pass-filtered, dynamic SSH from ECCO to show the fidelity of the ECCO solution by obtaining a North Atlantic SSH tripole pattern similar to observations (Fig. 1b, c).

The fidelity of the ECCO-V4r4 solution in simulating the oceanic heat fluxes was assessed by comparing the monthly MHT at 26°N and 41°N with existing observational estimates (Fig. 2a, b). The observational MHT estimates at these latitudes come from the RAPID-MOCHA project (https://mocha.earth.miami.edu/mocha/) and from the combination of satellite altimetry and Argo data[53,54]. The basin-wide and full-depth ocean heat content (H) change between 26°N and 41°N in the model is given by the following equation:

$$\frac{\partial H(t)}{\partial t} = \rho_0 c_p \left( \iint_{26N} v(x,z,t)\theta(x,z,t)dzdx - \iint_{41N} v(x,z,t)\theta(x,z,t)dzdx + \iiint F_{forc}^\theta(x,y,z,t)dV + \iiint F_{diff}^\theta(x,y,z,t)dV \right),$$
(1)

where $H(t) = \rho_0 c_p \iiint \theta(x,y,z,t)dV$ is the volume-integrated ocean heat content between 26°N and 41°N, $\rho_O$ is the reference density ($\rho_O = 1029\,kg\,m^{-3}$), $c_p$ is the specific heat capacity of seawater ($c_p = 3994\,J\,kg^{-1}\,°C^{-1}$), $\theta$ is potential temperature, $v$ is meridional velocity, $F_{forc}^\theta$ is total local forcing, which in the ECCO-V4r4 definition includes both surface and geothermal heat-flux, $F_{diff}^\theta$ symbolizes parameterized diffusive processes, and $V$ is the volume of the ocean between the two latitudes. The first two terms on the right side of (1) determine ocean heat convergence between 26°N and 41°N. The seasonal cycle was subtracted from the budget terms and the residual time series were low-pass filtered with a cutoff period of 1.5 years (Fig. 2c).

To investigate how the gyre-scale ocean variability affects coastal sea level and flood risk, we analyzed hourly records from 43 National Oceanic and Atmospheric Administration (NOAA) tide gauges along the U.S. east coast from Maine to Texas (Fig. 3a), available from NOAA's National Ocean Service (https://oceanservice.noaa.gov/). Sea level records are relative to the mean higher high water (MHHW) tidal datum, which is the average of the higher high water height of each tidal day observed over the National Tidal Datum Epoch (1983–2001). To account for the prevailing land subsidence at the U.S. tide gauges[46–49], we used vertical land motions based on the Glacial Isostatic Adjustment model ICE-5G v1.3 (ref. 58) distributed through the Permanent Service for Mean Sea Level (https://psmsl.org/). To determine the tripole-related sea level changes at tide gauges, the tide gauge records were regressed on the PC$_1$ of the interannual SSH variability. The amplitudes shown in Fig. 3a were computed as the half-range of the tripole-related sea level changes at tide gauges. We used the up-to-date minor flood threshold water levels published in a NOAA Technical Report[9]. A day was counted

as a flood day when the hourly averaged water level exceeded the minor flood threshold at least once in 24 h. The expected exceedances in Fig. 4 show the number of days per year, during which water levels exceeded a particular value above the MHHW. The relative contributions of the GMSL rise, the North Atlantic SSH tripole, and land subsidence on the frequency of floods were estimated by computing the number of annual flood days and expected exceedances after subtracting each of these processes from tide gauge records (Figs. 4 and 5).

## Data availability

All data used in this study is publicly available. The delayed-time satellite altimetry maps are distributed by the Copernicus Marine and Environment Monitoring Service (http://marine.copernicus.eu). EN.4.2.2 data are available from https://www.metoffice.gov.uk/hadobs/en4/ and are © British Crown Copyright, Met Office, provided under a non-commercial government license (http://www.nationalarchives.gov.uk/doc/non-commercial-government-licence/version/2/). The ECCO-V4r4 output is distributed by the Physical Oceanography Distributed Active Archive Center (PO.DAAC, https://podaac.jpl.nasa.gov/ECCO/). Data from the RAPID-MOCHA program are freely available at www.rapid.ac.uk/rapidmoc and www.mocha.earth.miami.edu/mocha. The MHT time series 41°N is available at https://zenodo.org/record/8170366. The tide gauge records are available from NOAA's National Ocean Service (https://oceanservice.noaa.gov/). The minor flood thresholds were published in a NOAA Technical Report[9] (https://oceanservice.noaa.gov/hazards/sealevelrise/sealevelrise-tech-report-sections.html). The rates of vertical land motions are available from the Permanent Service for Mean Sea Level (https://www.psmsl.org/).

## Code availability

The Matlab2022b was used for computations and plotting. Maps in Figs. 1 and 3 were plotted using M_Map package for Matlab[59], available online at www.eoas.ubc.ca/~rich/map.html. The analysis of the ECCO-V4r4 output was performed using the gcmfaces toolbox for Matlab freely available from https://gcmfaces.readthedocs.io/en/latest/. Codes to produce the figures are available from the corresponding author upon request.

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

## Acknowledgements

This research was carried out in part under the auspices of the Cooperative Institute for Marine and Atmospheric Studies, a cooperative institute of the University of Miami and NOAA (cooperative agreement number NA20OAR4320472), and at the Jet Propulsion Laboratory, California Institute of Technology under a contract from NASA. D.L.V., K.Z., and M.G. were supported by the National Oceanic and Atmospheric Administration (NOAA) Climate Variability and Predictability program (grant number NA20OAR4310407) and by the NOAA Atlantic Oceanographic and Meteorological Laboratory. H.Z. and D.M. were supported by NASA Modeling, Analysis, and Prediction (MAP) and Physical Oceanography (PO) programs.

## Author contributions

D.L.V. conceived the study, conducted data analysis, and wrote the paper. K.Z. computed ocean heat budget from the ECCO-V4r4 solution and contributed to data analysis. W.E.J. computed MHT at 26.5°N (RAPID array), and J.K.W. and W.H. computed MHT at 41°N. W.E.J., J.K.W., W.H., and M.G. contributed to the interpretation of the results. H.Z. and D.M. helped with the analysis of the ECCO-V4r4 output. D.L.V., K.Z., W.E.J., J.K.W., W.H., M.G., H.Z., D.M. shared the ideas and contributed to the preparation of the manuscript.

## Competing interests

The authors declare no competing interests.
