## [Peer Review File · Nature Communications]

Atlantic meridional overturning circulation increases flood risk along the United States southeast coastREVIEWER COMMENTS

Reviewer #1 (Remarks to the Author):

General comments:

1. The study addresses an important and timely issue that has been the focus of several recent papers- the unusual recent (since ~2010) acceleration in SLR on the South-Eastern and the Gulf coasts of the US. Past studies already pointed to various offshore atmospheric and oceanic (e.g., Gulf Stream weakening) sources for this regional SLR (Wdowinski et al., 2016; Valle-Levinson et al., 2017; Domingues et al., 2018, Ezer, 2019; some of them not cited). Here however, the authors found relation between coastal sea level and variations in AMOC and heat convergence around the subtropical gyre. The results are quite convincing, and the study also addresses the practical implications for increased coastal flooding. The approach and the results are clearly presented and the study worth publication.

However, some clarifications and more in-depth explanation of the mechanism involved are needed. Finding correlations between AMOC, the subtropical gyre heat conversion and sea level do not fully explain the underlying mechanism- for example, are the changes due to atmospheric wind pattern change (NAO?), air-sea interaction or changes in the deep-water formation that influence the strength of AMOC? Another missing point is that comparing sea level at different locations cannot ignore the contribution from vertical land movement (VLM); some of the stations experience significant land subsidence (what is the contribution of VLM compared with GMSL and the Tripole mode in Fig. 4?). The different response north and south of Cape Hatteras was also not fully explained (e.g., see Ezer, 2019, for a suggested mechanism that can support your results).

Specific comments and suggestions:

2. Line 23: "While the slowdown has not been observed to date" is not a completely accurate statement. It is true that the RAPID observations are relatively short to infer long-term (decadal) trends, but there are other observations and proxy data that show slowdown as well as some studies based on the RAPID data set itself (Ezer, 2015; Rahmstorf et al.,

2015; Jackson et al., 2016; Smeed et al., 2018). Please modify the statement and consider citing those studies in the introduction.

3. Lines 36-37: "...sea level fluctuations, driven by atmospheric pressure and wind forcing...", may add "such as tropical storms and hurricanes".

4. Lines 44-53: It is not so clear that what you describe is a recent shift in the "hotspot" of SLR from the area north of Cape Hatteras to south of Cape Hatteras, and why this occurred. You may add here a recent study (Ezer, 2019) that addressed this very point and suggests that the Gulf Stream separation from the coast can result in opposite sea level response north and south of CH when offshore conditions change. The mechanism suggested in this study can support the results here – when warmer waters injected into the GS, sea level would rise south of CH due to thermosteric effects when the GS is close to the coast, but after the GS separation increased temperature and sea surface height gradients across the GS will speed up the flow and reduce coastal sea level in the Mid-Atlantic Bight.

5. P. 2-5: Results section should be divided into sub sections for easier read.

6. Line 111: "As the tripole is strongly related to the OHC, its variability is partly driven by heat advection". What role do changes in wind pattern play here?

7. Line 120 and Fig. 2a: The heat transport at 41N shows a clear downward trend until ~2013 and an upward trend after- it will be interesting to know the value of these trends and their statistical significance. Since heat transport at 26N shows little or no trend, looking only at the difference does not tell the whole story why the changes occurred in the higher latitudes and their sources (maybe wind shift?).

8. Lines 156-159: "The amplitudes of the tripole-related coastal sea level changes... are small (0-2 cm) at the tide gauges situated to the north of Cape Hatteras, but they sometimes exceed 10 cm at the tide gauges located to the south of Cape Hatteras... ". See comment #4 and the study that contributes this result to the separation of the GS at CH.

9. Lines 181-200 and Fig. 4: This figure showing the contribution of GMSL vs. Tripole is great. However, first, it will be helpful to use the same vertical axes, and second, it will be advised to acknowledge and add the contribution of VLM. Land subsidence is a major part of large SLR and flooding in some of these locations (especially in the MAB and parts of the GOM). The introduction and methods also neglect to mention the role of VLM, which has been described in several studies that could be cited (Boon et al., 2010; Kolker et al., 2011; Karegar et al., 2016; Bekaert et al., 2017).

10. Line 283: "EOF1 explains 28% of the interannual SSH variance", this is not much- are higher EOFs insignificant compared to EOF1?

References

Bekaert DPS, Hamlington BD, Buzzanga B, Jones CE (2017). Spaceborne synthetic aperture radar survey of subsidence in Hampton Roads, Virginia (USA). *Sci Rep* 7:14752.

Doi:10.1038/s41598-017-15309-5.

Boon JD, Brubaker JM, Forrest DR (2010). Chesapeake Bay land subsidence and sea level change: an evaluation of past and present trends and future outlook. Special report in applied marine science and ocean engineering, 425, Virginia Institute of Marine Science.

Doi:10.21220/V58X4P.

Ezer, T., (2019). Regional differences in sea level rise between the Mid-Atlantic Bight and the South Atlantic Bight: Is the Gulf Stream to blame?, *Earth's Future*, 7(7), 771-783,

doi:10.1029/2019EF001174.

Jackson, L. C., Peterson, K. A., Roberts, C. D., & Wood, R. A. (2016). Recent slowing of Atlantic overturning circulation as a recovery from earlier strengthening. *Nature Geoscience*, 9(7), 518–522. Doi:10.1038/ngeo2715.

Karegar, M. A., Dixon, T. H., Engelhart, S. E. (2016). Subsidence along the Atlantic Coast of North America: insights from GPS and late Holocene relative sea level data. *Geophys Res*

Lett 43:3126–3133. doi:10.1002/2016GL068015.

Kolker, A. S., Allison, MA, Hameed, S. (2011). An evaluation of subsidence rates and sea-level variability in the northern Gulf of Mexico. *Geophys Res Lett* 38(21).

Doi:10.1029/2011GL049458.

Rahmstorf, S., Box, J. E., Feulner, G., Mann, M. E., Robinson, A., Rutherford, S., & Schaffernicht, E. J. (2015). Exceptional twentieth-century slowdown in Atlantic Ocean overturning circulation. *Nature Climate Change*, 5(5), 475–480. Doi:10.1038/nclimate2554.

Smeed, D. A., et al. (2018). The North Atlantic Ocean is in a state of reduced overturning. *Geophysical Research Letters*, 45(3), 1527-1533. doi:10.1002/2017GL076350.

Wdowinski, S., Bray, R., Kirtman, B. P., & Wu, Z. (2016). Increasing flooding hazard in coastal communities due to rising sea level: Case study of Miami Beach, Florida. *Ocean & Coastal Management*, 126, 1–8. Doi:10.1016/j.ocecoaman.2016.03.0.

Reviewer #2 (Remarks to the Author):

This manuscript uses observational datasets and ECCO state estimates to make two general conclusions: 1) large-scale heat transport convergence into the subtropical gyre (associated with the SSH/SST tripole) controls coastal sea level variability along the US east coast south of Cape Hatteras (including the gulf coast) and 2) that interannual variability in (dynamic) sea level has modulated the number of minor flood days as strongly as long-term changes in global mean sea level.

Since this is a short-form paper, it would seem simpler to have one message. In my assessment, the clear novelty is in the second message. With respect to the first message (which consumes substantial sections of methods, figures 1 and 2, and associated text), I'm concerned with the degree of overlap between this manuscript and Volkov et al. (2019).

Volkov, D. L., Lee, S.-K., Domingues, R., Zhang, H., & Goes, M. (2019). Interannual sea level

variability along the southeastern seaboard of the United States in relation to the gyre-scale heat divergence in the North Atlantic. *Geophysical Research Letters*, 46, 7481–7490.
<https://doi.org/10.1029/2019GL083596>

If more of the first message was referenced out of this paper, there is more space that could be devoted to implications on flooding, and its spatial variability. Some questions that came to my mind: Why do results differ at smaller scales (e.g. individual tide gauge, and groups of tide gauges, such as the Florida Gulf coast), both in terms of flood contribution and regression coefficient? Is there any relationship between the SSH/SST tripole and higher-than-interannual frequency water levels (e.g. storminess/IB/intrinsic ocean variability/eddies)? Can we say more than lines 260-261 with respect to predictability? What is driving the dramatic excursion in PC1 from 2010-2015 compared to the much more damped variability in PC1/heat transport convergence before ~2010.

In my reading, this paper, and Volkov et al. (2019), never differentiate the contributions of anomalous temperature and velocity to AMOC-associated OHT convergence. This makes the terms "AMOC-associated"/"AMOC-driven" ambiguous. As I read it the authors are always referring to heat transport and heat transport convergence. But I think of AMOC in terms of mass transport/volume transport. In the introduction (55-58), long-term changes/slowdown are in volume transport; I am not sure whether the interannual heat transport variations discussed here are analogous? It is worth at least clarifying terminology if this remains a significant part of this paper.

More minor points below, with line numbers:

24: "a secular slowdown"

25: intrinsic -- this term is confusing here as I think these changes are probably forced by the atmosphere...in anyway, they are not shown to be intrinsic here. I think the authors mean "natural" as in non-anthropogenic, but that is also not shown here.

54-55: not strictly true b/c of IB -- maybe be a little more precise: see Gregory et al. (2019) *Surveys in Geophysics*.

118: why look at correlations on a monthly basis if we're more concerned with the

interannual relationships with SSH (which is 1.5 year LP filtered)?

Fig 2c/2d: Are ECCO and observations temporally smoothed in some way, or is this just the effect of taking a large spatial average?

148: I'm not sure what the correlations are between here. Latitudes or MHT/AMOC?

162: Perhaps more important from a societal standpoint is the rate?

164-212: this is good, novel, and clear, if limited (e.g. only one threshold, no timeseries information, no exploration of the differences between locations, etc)

Fig 3a: a measure of regression significance would be helpful.

We thank both reviewers for their time and for the feedback they provided. Our answers to reviewers' questions and comments are highlighted by blue color below.

Reviewer #1 (Remarks to the Author):

General comments:

1. The study addresses an important and timely issue that has been the focus of several recent papers- the unusual recent (since ~2010) acceleration in SLR on the South-Eastern and the Gulf coasts of the US. Past studies already pointed to various offshore atmospheric and oceanic (e.g., Gulf Stream weakening) sources for this regional SLR (Wdowski et al., 2016; Valle-Levinson et al., 2017; Domingues et al., 2018, Ezer, 2019; some of them not cited). Here however, the authors found relation between coastal sea level and variations in AMOC and heat convergence around the subtropical gyre. The results are quite convincing, and the study also addresses the practical implications for increased coastal flooding. The approach and the results are clearly presented and the study worth publication.

We thank the reviewer for the good words. The citations for Wdowski et al. (2016) and Ezer (2019) that were missing in the original version have been added to the revised manuscript.

However, some clarifications and more in-depth explanation of the mechanism involved are needed. Finding correlations between AMOC, the subtropical gyre heat conversion and sea level do not fully explain the underlying mechanism- for example, are the changes due to atmospheric wind pattern change (NAO?), air-sea interaction or changes in the deep-water formation that influence the strength of AMOC?

While we agree that the questions raised by the reviewer about the underlying mechanisms for the AMOC variability are important and interesting, we would like to stress that the objective of this study was to assess how the gyre-scale dynamic SSH changes in the subtropical North Atlantic, partly driven by the AMOC-induced heat convergence, influence the frequency of coastal inundations. The question of why the AMOC varies is not a simple question that can be addressed in one paper, and it clearly goes beyond the scope of our study. There have been many observational and modeling studies addressing the variability of the AMOC on different time scales and at different locations, and the mechanistic understanding of the AMOC changes is still far from being complete. In this study, we just assume that the AMOC varies. Nevertheless, using the ECCO state estimate, we quantify the relative contributions of the net surface heat flux and advection to the variability of oceanic heat content in the North Atlantic subtropical gyre. To address the reviewer's comment on the atmospheric wind pattern changes, the revised version of the manuscript now mentions in lines 93-95 that the North Atlantic SSH tripole is correlated with the NAO, as has been shown in Volkov et al. (2019).

Another missing point is that comparing sea level at different locations cannot ignore the contribution from vertical land movement (VLM); some of the stations experience significant land subsidence (what is the contribution of VLM compared with GMSL and the Tripole mode in Fig. 4?).

The VLM is indeed an important contributor to coastal sea level changes. In the Introduction of the original version of the manuscript, we listed the VLM as one of the processes responsible for the regional acceleration of SLR along the U.S. east coast, but we decided not to consider it in the following analysis and keep the focus on the contributions of the North Atlantic SSH tripole and the GMSL rise. Per Reviewer's request, we included the contribution of the VLM in Fig. 4 of the revised manuscript. Relevant discussion has also been added to the text (in the Introduction, lines 66-68; in Section 2.4., lines 198-206; and in Methods, lines 350-353), although we tried to limit the discussion on VLM since this is an external factor and not the main focus of this research.

The different response north and south of Cape Hatteras was also not fully explained (e.g., see Ezer, 2019, for a suggested mechanism that can support your results).

In the Introduction of the revised manuscript, we have added a sentence referring to Ezer (2019) (lines 48-49).

Specific comments and suggestions:

2. Line 23: "While the slowdown has not been observed to date" is not a completely accurate statement. It is true that the RAPID observations are relatively short to infer long-term (decadal) trends, but there are other observations and proxy data that show slowdown as well as some studies based on the RAPID data set itself (Ezer, 2015; Rahmstorf et al., 2015; Jackson et al., 2016; Smeed et al., 2018). Please modify the statement and consider citing those studies in the introduction.

We agree with the reviewer, and we have modified the statement as follows: "While century-long proxy data indicates that the AMOC may already be slowing down³⁹, the direct observations of the AMOC are still too short to confirm this centennial trend⁴⁰, and they mainly represent interannual-to-decadal variations⁴¹⁻⁴⁴." (Lines 62-64).

3. Lines 36-37: "...sea level fluctuations, driven by atmospheric pressure and wind forcing...", may add "such as tropical storms and hurricanes".

Thank you for this suggestion. We have modified the sentence accordingly.

4. Lines 44-53: It is not so clear that what you describe is a recent shift in the "hotspot" of SLR from the area north of Cape Hatteras to south of Cape Hatteras, and why this occurred. You may add here a recent study (Ezer, 2019) that addressed this very point and suggests that the Gulf Stream separation from the coast can result in opposite sea

level response north and south of CH when offshore conditions change. The mechanism suggested in this study can support the results here – when warmer waters injected into the GS, sea level would rise south of CH due to thermosteric effects when the GS is close to the coast, but after the GS separation increased temperature and sea surface height gradients across the GS will speed up the flow and reduce coastal sea level in the Mid-Atlantic Bight.

Thank you for the helpful remark. In lines 48-49, we have added the following sentence to the paragraph to clarify the statement about the shift in the hotspot of SLR in 2010: “A recent shift in the hotspot of the accelerated sea level rise that occurred in 2010 has been attributed to changes in the Gulf Stream strength and position (Ezer, 2019)”.

5. P. 2-5: Results section should be divided into sub sections for easier read.

We have subdivided the Results section into four subsections.

6. Line 111: “As the tripole is strongly related to the OHC, its variability is partly driven by heat advection”. What role do changes in wind pattern play here?

In the previous subsection (line 93), we have mentioned that the tripole is correlated with the NAO, but the details of the relationship between wind forcing and the tripole lie beyond the scope of this study.

7. Line 120 and Fig. 2a: The heat transport at 41N shows a clear downward trend until ~2013 and an upward trend after- it will be interesting to know the value of these trends and their statistical significance. Since heat transport at 26N shows little or no trend, looking only at the difference does not tell the whole story why the changes occurred in the higher latitudes and their sources (maybe wind shift?).

We agree that this is an interesting question. However, with this short manuscript we are not aiming and not able to answer the question of what led to the changes at higher latitudes. We prefer to leave this question for a dedicated study on the AMOC variability and keep the focus of the present study on the relationship between the AMOC-induced heat convergence in the North Atlantic subtropical gyre and the frequency of floods along the U.S. east coast.

8. Lines 156-159: “The amplitudes of the tripole-related coastal sea level changes... are small (0-2 cm) at the tide gauges situated to the north of Cape Hatteras, but they sometimes exceed 10 cm at the tide gauges located to the south of Cape Hatteras... “. See comment #4 and the study that contributes this result to the separation of the GS at CH.

While changes in the Gulf Stream strength and position around 2010 can affect the regional changes in coastal sea level, here we find that the amplitudes of the tripole-related changes during the entire study period (1993-2020) are small north of Cape Hatteras. We think that this is probably because tide gauges in this region are located

near the boundary between the subtropical and the subpolar bands of the tripole, where the amplitudes of the tripole-related changes are small. In lines 169-171, we include the following sentence: “The small amplitudes of the tripole-related SSH changes at the former tide gauges are probably due to the proximity to the boundary between the subtropical and the subpolar bands of the tripole.”

9. Lines 181-200 and Fig. 4: This figure showing the contribution of GMSL vs. Tripole is great. However, first, it will be helpful to use the same vertical axes, and second, it will be advised to acknowledge and add the contribution of VLM. Land subsidence is a major part of large SLR and flooding in some of these locations (especially in the MAB and parts of the GOM). The introduction and methods also neglect to mention the role of VLM, which has been described in several studies that could be cited (Boon et al., 2010; Kolker et al., 2011; Karegar et al., 2016; Bekaert et al., 2017).

We have followed the reviewer’s suggestion and included the impact of the VLM in Fig. 4 (green curves). We have also used the same vertical axes and added suggested references.

10. Line 283: “EOF1 explains 28% of the interannual SSH variance”, this is not much— are higher EOFs insignificant compared to EOF1?

The second mode is also significant, and it explains about 14% of the variance. We do not consider this mode, however, because it mostly contributes to the SSH variance in the northeastern part of the subpolar North Atlantic (Volkov et al., 2022; referenced in this study), while EOF1 is an important contributor to the local SSH variance in both the subpolar and the subtropical gyres, but most importantly also along the U.S. southeast and Gulf coasts. In lines 311-316, we have added the following text: “The EOF₁ mode explains the majority of the interannual SSH variance in the northwestern North Atlantic, in the subtropical gyre, and – most importantly for the objectives of this study – along the South Atlantic Bight, the Gulf of Mexico, and the Caribbean coasts³². The second EOF mode explains 14% of the interannual SSH variance, but it mostly accounts for the interannual SSH variability in the northeastern part of the subpolar North Atlantic⁴⁹. Therefore, this mode is not considered in this study.”

We would also like to note that 28% of the explained variance by EOF1 is a relatively large number given that it relates to the entire North Atlantic and that no spatial filtering was applied to the SSH data resulting in residual eddy-like features still seen in Fig. 1a.

References

Bekaert DPS, Hamlington BD, Buzzanga B, Jones CE (2017). Spaceborne synthetic aperture radar survey of subsidence in Hampton Roads, Virginia (USA). *Sci Rep* 7:14752. Doi:10.1038/s41598-017-15309-5.

Boon JD, Brubaker JM, Forrest DR (2010). Chesapeake Bay land subsidence and sea level change: an evaluation of past and present trends and future outlook. *Special*

report in applied marine science and ocean engineering, 425, Virginia Institute of Marine Science. Doi:10.21220/V58X4P.

Ezer, T., (2019). Regional differences in sea level rise between the Mid-Atlantic Bight and the South Atlantic Bight: Is the Gulf Stream to blame?, *Earth's Future*, 7(7), 771-783, doi:10.1029/2019EF001174.

Jackson, L. C., Peterson, K. A., Roberts, C. D., & Wood, R. A. (2016). Recent slowing of Atlantic overturning circulation as a recovery from earlier strengthening. *Nature Geoscience*, 9(7), 518–522. Doi:10.1038/ngeo2715.

Karegar, M. A., Dixon, T. H., Engelhart, S. E. (2016). Subsidence along the Atlantic Coast of North America: insights from GPS and late Holocene relative sea level data. *Geophys Res Lett* 43:3126–3133. doi:10.1002/2016GL068015.

Kolker, A. S., Allison, MA, Hameed, S. (2011). An evaluation of subsidence rates and sea-level variability in the northern Gulf of Mexico. *Geophys Res Lett* 38(21). Doi:10.1029/2011GL049458.

Rahmstorf, S., Box, J. E., Feulner, G., Mann, M. E., Robinson, A., Rutherford, S., & Schaffernicht, E. J. (2015). Exceptional twentieth-century slowdown in Atlantic Ocean overturning circulation. *Nature Climate Change*, 5(5), 475–480. Doi:10.1038/nclimate2554.

Smeed, D. A., et al. (2018). The North Atlantic Ocean is in a state of reduced overturning. *Geophysical Research Letters*, 45(3), 1527-1533. doi:10.1002/2017GL076350.

Wdowinski, S., Bray, R., Kirtman, B. P., & Wu, Z. (2016). Increasing flooding hazard in coastal communities due to rising sea level: Case study of Miami Beach, Florida. *Ocean & Coastal Management*, 126, 1–8. Doi:10.1016/j.ocecoaman.2016.03.0.

All these references have been added to the revised version of the manuscript.

Reviewer #2 (Remarks to the Author):

This manuscript uses observational datasets and ECCO state estimates to make two general conclusions: 1) large-scale heat transport convergence into the subtropical gyre (associated with the SSH/SST tripole) controls coastal sea level variability along the US east coast south of Cape Hatteras (including the gulf coast) and 2) that interannual variability in (dynamic) sea level has modulated the number of minor flood days as strongly as long-term changes in global mean sea level.

Since this is a short-form paper, it would seem simpler to have one message. In my assessment, the clear novelty is in the second message. With respect to the first

message (which consumes substantial sections of methods, figures 1 and 2, and associated text), I'm concerned with the degree of overlap between this manuscript and Volkov et al. (2019).

Volkov, D. L., Lee, S.-K., Domingues, R., Zhang, H., & Goes, M. (2019). Interannual sea level variability along the southeastern seaboard of the United States in relation to the gyre-scale heat divergence in the North Atlantic. *Geophysical Research Letters*, 46, 7481–7490. <https://doi.org/10.1029/2019GL083596>

We would say it somewhat differently: the novelty and the main message of this study is that the AMOC-induced heat convergence in the subtropical North Atlantic in 2015-2019 modulates the number of floods as strongly as the GMSL rise. Volkov et al. (2019) only showed that (i) the North Atlantic SSH tripole is correlated with sea level changes along the South Atlantic Bight and Gulf coasts and (ii) that the tripole is correlated with the NAO and meridional heat transport. In the present study, while focusing mainly on how the tripole impacts the frequency of floods, we also assess the full-depth regional heat budget, thus explicitly showing that heat convergence in the subtropical gyre is mainly driven by heat advection, and that the contribution of the net surface heat flux is of secondary importance.

If more of the first message was referenced out of this paper, there is more space that could be devoted to implications on flooding, and its spatial variability. Some questions that came to my mind: Why do results differ at smaller scales (e.g. individual tide gauge, and groups of tide gauges, such as the Florida Gulf coast), both in terms of flood contribution and regression coefficient? Is there any relationship between the SSH/SST tripole and higher-than-interannual frequency water levels (e.g. storminess/IB/intrinsic ocean variability/eddies)? Can we say more than lines 260-261 with respect to predictability? What is driving the dramatic excursion in PC1 from 2010-2015 compared to the much more damped variability in PC1/heat transport convergence before ~2010.

The reviewer lists several interesting questions, and each of them deserves a dedicated paper or even papers. As the reviewer justly pointed out above, this is a short-form paper, in which we wanted to highlight the importance of large-scale oceanic processes in modulating the frequency of coastal inundations and to link coastal sea level and inundations to the AMOC. Obviously, we are not able to cover all relevant details in one letter-type manuscript.

Regarding the differences at smaller scales, between individual tide gauges or their groups, one needs to consider local geography (coast configuration and bottom topography), local tides (including shallow water tides), vertical land motions (included in the revised version of the paper), local changes in atmospheric forcing, impact of eddies, Rossby and coastal trapped waves, etc., which are beyond the scope of this study. Nevertheless, some of these processes have already been addressed by earlier studies referenced in the manuscript.

Regarding the variability of the tripole (PC1), including the dramatic excursion of PC1 in 2010-2015, we do show that this variability is mostly driven by the AMOC-induced heat advection. The question of what drove the associated changes in the AMOC is beyond the scope of this study.

In my reading, this paper, and Volkov et al. (2019), never differentiate the contributions of anomalous temperature and velocity to AMOC-associated OHT convergence. This makes the terms "AMOC-associated"/"AMOC-driven" ambiguous. As I read it the authors are always referring to heat transport and heat transport convergence. But I think of AMOC in terms of mass transport/volume transport. In the introduction (55-58), long-term changes/slowdown are in volume transport; I am not sure whether the interannual heat transport variations discussed here are analogous? It is worth at least clarifying terminology if this remains a significant part of this paper.

This is a very good point, and we thank the reviewer for bringing it up. We have addressed it at the beginning of Section 2.3 of the revised manuscript (lines 155-161): "The MHT and the AMOC estimates are strongly correlated at 26°N and 41°N ($r = 0.96$ at 26°N and $r = 0.85$ at 41°N), meaning that the MHT variability is mainly due to the variability of the meridional velocity rather than temperature. Furthermore, the MHT variability at the two locations is dominated by the overturning circulation and not by the horizontal gyre circulation^{52,53}. Therefore, the MHT convergence anomalies between the two latitudes mostly denote the AMOC-driven OHC tendencies (warming and cooling) in the subtropical band of the tripole (red and blue shading in Fig. 2d)."

More minor points below, with line numbers:

24: "a secular slowdown"

We have modified the beginning of the sentence as follows: "While direct observations of the AMOC are still too short to infer long-term trends..." (lines 22-23).

25: intrinsic -- this term is confusing here as I think these changes are probably forced by the atmosphere...in anyway, they are not shown to be intrinsic here. I think the authors mean "natural" as in non-anthropogenic, but that is also not shown here.

We agree with the reviewer and remove the word "intrinsic".

54-55: not strictly true b/c of IB -- maybe be a little more precise: see Gregory et al. (2019) Surveys in Geophysics.

We thank the reviewer for this remark. The sentence was edited to include a statement that the regional sea level variations have been corrected for the IB effect (lines 56-57).

118: why look at correlations on a monthly basis if we're more concerned with the interannual relationships with SSH (which is 1.5 year LP filtered)?

This comparison is shown only to demonstrate the robustness of the MHT in the model. We think using the monthly time series is better, because they capture some individual events that are smoothed out with a low-pass filtering.

Fig 2c/2d: Are ECCO and observations temporally smoothed in some way, or is this just the effect of taking a large spatial average?

Yes, the time series were smoothed with a low-pass filter with the cutoff period of 1.5 years. This is mentioned in lines 342-344 (Methods).

148: I'm not sure what the correlations are between here. Latitudes or MHT/AMOC?

We have modified this sentence as follows (lines 155-156): "The MHT and the AMOC estimates are strongly correlated at 26°N and 41°N ($r = 0.96$ at 26°N and $r = 0.85$ at 41°N)..."

162: Perhaps more important from a societal standpoint is the rate?

The sentence the reviewer is referring to reads: "*This means that along the U. S. Southeast and Gulf coasts the impact of the tripole, i.e., the gyre-scale dynamic SSH changes, on the frequency of coastal inundations is equal or sometimes even greater than the impact of the GMSL rise.*" If the reviewer is referring to the total rate composed of the gyre-scale dynamic SSH change and the GMSL rise, then definitely this rate is one of the most important characteristics, but in the present context, we are interested in the frequency of inundations caused by the dynamic sea level changes only.

164-212: this is good, novel, and clear, if limited (e.g. only one threshold, no timeseries information, no exploration of the differences between locations, etc)

Fig 3a: a measure of regression significance would be helpful.

Thank you. As we have stated earlier, this is a short-form paper. Hence, we are not able to cover all aspects of regional sea level changes. Regarding the time series information, please note that tide gauge records are quite busy, because they include tides, so there is not much visual benefit of including a time series like this. The 95% confidence interval for regression is shown in the revised Fig. 3 (insert).

REVIEWERS' COMMENTS

Reviewer #1 (Remarks to the Author):

The authors responded sufficiently to the reviewers' comments so the revised manuscript is much better now. I only have one further suggestion before accepting the paper: a recent paper (also published in Nature Communications) by Dangendorf et al., 2023 (<https://www.nature.com/articles/s41467-023-37649-9>) deals with similar issue of accelerated sea level and flooding in the southeast and Gulf coasts due to remote ocean dynamics. So I suggest the authors look at this paper and cite it to give this topic more attention.

Reviewer #2 (Remarks to the Author):

I am satisfied with the authors' revisions, and hope that they pursue some of my additional suggestions in future work.

Response to the reviewers' comments

We thank both reviewers for their constructive and insightful comments that helped us to improve the manuscript. Our responses are highlighted by the blue color.

Reviewer #1 (Remarks to the Author):

The authors responded sufficiently to the reviewers' comments so the revised manuscript is much better now. I only have one further suggestion before accepting the paper: a recent paper (also published in Nature Communications) by Dangendorf et al., 2023 (<https://www.nature.com/articles/s41467-023-37649-9>) deals with similar issue of accelerated sea level and flooding in the southeast and Gulf coasts due to remote ocean dynamics. So I suggest the authors look at this paper and cite it to give this topic more attention.

We thank the reviewer for bringing a recent paper by Dangendorf et al (2023) to our attention. We have included a reference to this paper in the Introduction (line 56): "This regional acceleration and spatial variation of sea level rise has been attributed to different processes, such as ..., **the combined influence of external forcing and internal climate variability**³⁴, ...".

Reviewer #2 (Remarks to the Author):

I am satisfied with the authors' revisions, and hope that they pursue some of my additional suggestions in future work.

We thank the reviewer for the helpful suggestions that we will definitely consider in a follow-up research.